# Multi-Source Causal Inference Using Control Variates under Outcome Selection Bias

**Wenshuo Guo**[*]                                                          *wguo@cs.berkeley.edu*
*Department of Electrical Engineering and Computer Sciences*
*University of California, Berkeley*

**Serena Wang**[*]                                                          *serenalwang@berkeley.edu*
*Department of Electrical Engineering and Computer Sciences*
*University of California, Berkeley*

**Peng Ding**                                                          *pengdingpku@berkeley.edu*
*Department of Statistics*
*University of California, Berkeley*

**Yixin Wang**                                                          *yixinw@umich.edu*
*Department of Statistics*
*University of Michigan*

**Michael I. Jordan**                                                          *jordan@cs.berkeley.edu*
*Department of Electrical Engineering and Computer Sciences, and Department of Statistics*
*University of California, Berkeley*

**Reviewed on OpenReview:** *https: // openreview. net/ forum? id= CrimIjBa64*

## Abstract

While many areas of machine learning have benefited from the increasing availability of large and varied datasets, the benefit to causal inference has been limited given the strong assumptions needed to ensure the identifiability of causal effects —which are often not satisfied in real-world datasets. For example, many large observational datasets (e.g., case-control studies in epidemiology, click-through data in recommender systems) suffer from selection bias on the outcome, which makes the average treatment effect (ATE) non-identifiable. We propose an algorithm to estimate causal effects from *multiple* data sources, where the ATE may be identifiable only in some datasets but not others. The idea is to construct control variates across the datasets in which the ATE may not be identifiable, which provably reduces the variance of the ATE estimate. We focus on a setting where the observational datasets suffer from outcome selection bias, assuming access to an auxiliary small dataset from which we can obtain a consistent estimate of the ATE. We propose a construction of control variate by taking the difference of the conditional odds ratio estimates from the two datasets. Across simulations and two case studies with real data, we show that the control variate-based ATE estimator has consistently and significantly reduced variance against different baselines.

## 1 Introduction

The ongoing rapid growth in the scale and scope of data sources has challenged many research communities, including optimization, machine learning, and causal inference (Hand, 2007; Agarwal & Duchi, 2012; Bottou et al., 2018; Shiffrin, 2016). In particular, in causal inference, there has been a surge of interest in developing tools to draw causal conclusions from large-scale observational data (Imbens & Rubin, 2015; Pearl, 2009;

---

[*]Equal contribution.

Maathuis et al., 2010; Kleinberg, 2013; Maslove & Leisman, 2019; Wachinger et al., 2019). Compared to randomized trial designs, these observational data sources can often offer longitudinal data, fine-grained measurements, and much larger sample sizes. For example, electronic health data, including electronic health records (EHR) used for clinical care, may contain extensive details that include the timing, intensity, and quality of the interventions received by individuals. Moreover, randomized clinical trials are sometimes not feasible due to logistical, economic, or ethical reasons, and even when feasible can be seriously limited in terms of sample size (Stuart et al., 2013). Thus, large-scale observational datasets hold open the promise of a much greater impact for causal inference methodology.

However, conceptual problems arise in the observational data setting which can make causal effects unidentifiable or hard to estimate. The problems include unmeasured confounding, noisy measurements, inconsistency, and selection bias (Rosenbaum & Rubin, 1983; Angrist et al., 1996; Nalatore et al., 2007; Hernán et al., 2004). These problems have generally been studied in the setting of a single observational data source, and in such a setting it is natural to view them through an all-or-nothing lens—either selection bias is present or it is not, either confounding is present or it is not, etc. In such cases, causal inference is possible only when the data source satisfies certain delicate assumptions. These assumptions are often invalid; in particular, selection bias is notorious for being difficult to assume away—for example, in case-control datasets in epidemiological studies, cases are much more likely to be reported than non-cases; in observational data in recommender systems, certain items are more likely to receive clicks and ratings (Rothman et al., 2008; Robins et al., 2000; Robins, 2001; Hernán et al., 2004; Wang et al., 2016; Schnabel et al., 2016; Wang et al., 2020). Under the existence of such selection bias, causal effects are in general unidentifiable (Correa et al., 2018; 2019).

In this paper, we take a different route and consider estimating causal effects from *multiple* data sources. Can we combine large, possibly biased datasets with smaller, unbiased datasets to develop efficient estimators of causal effects? Our work is motivated by the observation that, in practice, we are often able to obtain a small dataset where the causal effects are identifiable; e.g., from small-scale randomized trials or observational data with limited known confounding. Causal inference may not be efficient in such small datasets alone due to limited sample size. However, the large observational datasets, while not permitting causal inference by themselves, may be useful in improving the efficiency of the causal effects estimators from the small unbiased dataset.

We present an affirmative answer to this question in this paper. We show that, by leveraging carefully constructed control variates, one can perform causal inference when the average treatment effect (ATE) may be identifiable only in some datasets but not others. Though control variate is a classical technique, the difficulty in applying it to causal inference lies in the construction of valid control variates, which requires one to find aspects of the data that are simultaneously correlated with ATE–even when the ATE is non-identifiable—and transportable across datasets. Such control variates allow us to design new ATE estimators which enjoy variance reduction, which we theoretically quantify.

To apply control variates to multi-source causal inference, we focus on a setting where some data sources suffer from outcome selection bias, which is prevalent in case-control studies. Outcome selection bias renders ATE non-identifiable and challenges causal inference. To combine such datasets for causal inference, we propose to form control variates using the odds ratio estimates across datasets; odds ratio is identifiable even in the presence of outcome selection bias. We establish the theoretical validity of this control variate construction. We also show empirically that this control variate can significantly reduce the variance of ATE estimates using a variety of estimators across synthetic data and two real-data case studies.

## 1.1 Related work

The problem of combining multiple datasets to estimate causal effects has attracted much recent research interest given the strong practical incentives, especially combining datasets from observational and experimental sources (Colnet et al., 2020; Rosenfeld et al., 2017; Rosenman et al., 2018; 2020; Kallus et al., 2018; Triantafillou et al., 2021), where the observational data may suffer from hidden confounders and complex patterns of missing data. Yang & Ding (2020) propose estimators by combining a main dataset with unmeasured confounders and a smaller validation dataset with supplementary information on these confounders; Cannings & Fan (2019) propose new estimators by combining datasets with complete cases and further

observations with missing values, in order to improve on the performance of the complete-case estimator; they construct multiple error-prone estimators that are transportable across the main and validation datasets. In fact, their "error-prone estimators" are used to design one particular choice of control variates in our framework. However, their estimators rely on identifiability of the ATE in observational data, and therefore do not handle selection bias. Finally, Correa et al. (2019) consider identifiability when given access to a biased data distribution and an unbiased data distribution, and propose an algorithm for combining these. However, their work only considers full data distributions and does not propose finite sample estimators, which are the focus of this work.

Selection bias is induced by preferential selection of data points, and it is often governed by unknown factors that can interact with treatments, outcomes and their consequences. Operationally, selection bias cannot be easily eliminated by random sampling. There have been extensive studies on methods that deal with mitigating certain selection biases in observational studies. Bareinboim & Pearl (2012) discuss graphical and algebraic methods, and derive a general condition together with a procedure for recovering the odds ratio under selection bias. They also propose using instrumental variables for the removal of selection bias in the presence of confounding bias. Zhang (2008) studies special cases in which selection bias can be detected even from the observations, as captured by a non-chordal undirected graphical component. Robins et al. (2000) and Hernán et al. (2004) propose epidemiological methods that assume knowledge of the probability of selection given treatment, which can be estimated from data in certain cases.

The control variates technique is a classical tool for variance reduction, and there have been applications of it to causal effect estimation (Tan, 2006). Here we use the control variates technique for variance reduction in multi-source causal inference. The key technical development of our approach is the design of a valid control variate for multi-source causal inference. To this end, we propose to identify an estimand that is transportable between the observational data—i.e., the selection-biased dataset—to the experimental data. If this estimand has sufficient correlation with the target estimand of interest, it can be used to construct control variates. To this end, our work relates to the literature on transportability in causal inference (Bareinboim & Pearl, 2014; Lee et al., 2020; Bareinboim & Pearl, 2016). These works investigate what causal quantities are identifiable, which suggests potential ways to construct control variates.

## 2 Preliminaries

In this section, we first present the basic setup for causal inference with multiple data sources. We then formalize our assumptions on the identification of causal effects.

**Potential outcomes and ATE estimation.** We use the potential outcomes framework to define causal effects (Neyman, 1923; Rubin, 1974). Let $Z$ denote a binary treatment random variable, with 0 and 1 being the labels for control and active treatments, respectively. For each realization of the level of treatment $z \in \{0, 1\}$, we assume that there exists a potential outcome $Y(z)$ representing the outcome had the subject been given treatment $z$ (possibly contrary to fact). Then, the observed outcome is $Y = Y(Z) = ZY(1) + (1 - Z)Y(0)$. Further, we denote a vector of observed pretreatment covariates as $X \in \mathcal{X} \subseteq \mathbb{R}^d$. We focus on estimating the ATE: $\tau = E[Y(1) - Y(0)]$.

**Data sources.** We consider a main data source that consists of observations $\mathcal{O}_1 = \{(Z_i, X_i, Y_i) : i \in \mathcal{S}_1\}$, with sample size $n_1 = |\mathcal{S}_1|$, and a validation data source with observations $\mathcal{O}_2 = \{(Z_j, X_j, Y_j) : j \in \mathcal{S}_2\}$ and sample size $n_2 = |\mathcal{S}_2|$. We assume that the ATE is identifiable only from the validation data source but not the main data source, and generally $n_2 < n_1$. For simplicity, we consider two data sources, but generalizing to multiple is straightforward.

A fundamental problem in causal inference is that the counterfactuals are not observable. Therefore, to allow for the identification of ATE, we make the following ignorability assumption (Rosenbaum & Rubin, 1983) with respect to the validation data $\mathcal{O}_2$.

**Assumption 2.1** (Ignorability)**.** $Y(z) \perp\!\!\!\perp Z \mid X$ *for* $z = 0, 1$.

Under Assumption 2.1, many methods for estimating the ATE from a single observational dataset exist in the causal inference literature (see, e.g., Rosenbaum, 2002; Imbens, 2004; Rubin, 2006).

## 3    A General Strategy with Control Variates

We first outline a general strategy for efficient estimation of the ATE by utilizing both the main and validation data. Such a strategy allows us to design efficient ATE estimators using all the data, without requiring the ATE to be identifiable in all individual data sources. Informally, we want to identify features that are transportable across both datasets and robust to the type of confounding or bias affecting the main data. Using such features, we exploit information across the datasets and improve the efficiency of the ATE estimator using all datasets. This strategy is reminiscent of a control-variate methodology for general variance reduction in Monte Carlo simulations (Owen, 2013).

Let $\psi \in \mathbb{R}^m$ be an estimand for which there exist consistent estimators obtainable from datasets $\mathcal{O}_1$ and $\mathcal{O}_2$ (with consistent estimators denoted by $\widehat{\psi}_1$ and $\widehat{\psi}_2$, respectively). The key requirement of *transportability* is that $\widehat{\psi}_1 - \widehat{\psi}_2$ converges asymptotically to zero. Let $\widehat{\tau}_2$ denote a consistent estimator of the true ATE $\tau$ that we obtain using dataset $\mathcal{O}_2$ with asymptotic variance $v_2$. In particular, we consider a class of estimators satisfying

$$n_2^{1/2} \left( \begin{array}{c} \widehat{\tau}_2 - \tau \\ \widehat{\psi}_2 - \widehat{\psi}_1 \end{array} \right) \to \mathcal{N} \left\{ 0, \left( \begin{array}{cc} v_2 & \Gamma^\top \\ \Gamma & V \end{array} \right) \right\}, \tag{1}$$

for some $V \in \mathbb{R}^{m \times m}$ and $\Gamma \in \mathbb{R}^{m \times 1}$. If Eq. (1) holds exactly rather than asymptotically, by multivariate normal theory, we have the following the conditional distribution:

$$n_2^{1/2}(\widehat{\tau}_2 - \tau) \mid n_2^{1/2}(\widehat{\psi}_2 - \widehat{\psi}_1) \sim \mathcal{N} \left\{ n_2^{1/2} \Gamma^\top V^{-1}(\widehat{\psi}_2 - \widehat{\psi}_1), v_2 - \Gamma^\top V^{-1}\Gamma \right\}.$$

We apply the method of control variates (Owen, 2013) by using the estimators for $\tau$ and $\psi$ jointly to build a new estimator of $\tau$ which has a lower variance than $\hat{\tau}_2$. Specifically, we construct a new estimator for ATE using control variates as follows: $\widehat{\tau}_{\mathrm{CV}}(\beta) = \widehat{\tau}_2 - \beta^\top(\widehat{\psi}_2 - \widehat{\psi}_1)$. Solving for the optimal $\beta$, we obtain the new estimator

$$\widehat{\tau}_{\mathrm{CV}} = \widehat{\tau}_2 - \Gamma^\top V^{-1}(\widehat{\psi}_2 - \widehat{\psi}_1), \tag{2}$$

where $V = \mathrm{Var}(\widehat{\psi}_2 - \widehat{\psi}_1)^{-1}$ and $\Gamma = \mathrm{Cov}(\widehat{\psi}_2 - \widehat{\psi}_1, \widehat{\tau}_2)$.

**Theorem 3.1.** *(Owen, 2013; Yang & Ding, 2020) Denote the asymptotic variance of $\widehat{\tau}_2$ as $v_2$. Under Assumption 2.1, if Eq. 1 holds, then $\widehat{\tau}_{CV}$ is consistent for $\tau$, and we have: $n_2^{1/2}(\widehat{\tau}_{CV} - \tau) \to \mathcal{N}(0, v_2 - \Gamma^\top V^{-1}\Gamma)$, in distribution as $n_2 \to \infty$ with ratio $n_2/n_1 \to \rho \in [0, 1]$ converging to a constant. Given a nonzero $\Gamma$, the asymptotic variance, $v_2 - \Gamma^\top V^{-1}\Gamma$, is smaller than $v_2$.*

From a practical standpoint, Theorem 3.1 shows that the most effective control variate estimators $\widehat{\psi}_1 - \widehat{\psi}_2$ will have low variance and high correlation with the ATE estimator $\widehat{\tau}_2$. Moreover, the key to use such a method is to find control variate estimators $\widehat{\psi}_1 - \widehat{\psi}_2$ that are consistent across the datasets. Once such control variate estimators are constructed, we do not any further assumptions across the datasets, such as the two datasets must share exactly the same causal structure model. Empirically, to estimate the optimal value of $\beta$, we can use estimators $\widehat{V}$ and $\widehat{\Gamma}$ for the variance and covariance in Eq. 2. These estimators $\widehat{V}$ and $\widehat{\Gamma}$ can be obtained by bootstrap sampling, with details in Appendix B.

Note that here we illustrated the main approach with one single observation dataset $\mathcal{O}_1$ and one validation dataset $\mathcal{O}_2$. However, it is straightforward to extend to multiple observational data sources. To see that, suppose that there is one validation dataset $\mathcal{O}_{val}$, and multiple observational datasets $\mathcal{O}_{obs}^1, \cdots, \mathcal{O}_{obs}^K$. Let $\psi^j \in \mathbb{R}^m$ be an estimand for which there exist consistent estimators obtainable from datasets $\mathcal{O}_{val}$ and $\mathcal{O}_{obs}^j$ for $j = 1, \cdots K$ (with consistent estimators denoted by $\widehat{\psi}_{val}^j$ and $\widehat{\psi}_{obs}^j$, respectively). Then, for each $j \in [K]$, $\widehat{\psi}_{val}^j - \widehat{\psi}_{obs}^j$ converges asymptotically to zero. Therefore, we can define the control variates to be a stacked vector:

$$\widehat{\mathrm{CV}} = \left( \begin{array}{c} \widehat{\phi}_{obs}^1 - \widehat{\phi}_{val}^1 \\ \cdots \\ \widehat{\phi}_{obs}^K - \widehat{\phi}_{val}^K \end{array} \right)$$

Note that this stacked vector of control variates still satisfy that each entry of it converges asymptotically to zero. Thus, we have

$$n_2^{1/2} \begin{pmatrix} \widehat{\tau}_2 - \tau \\ \widehat{\mathrm{CV}} \end{pmatrix} \to \mathcal{N} \left\{ 0, \begin{pmatrix} v_2 & \Gamma^\top \\ \Gamma & V \end{pmatrix} \right\}, \tag{3}$$

and Theorem 3.1 applies.

## 4 Control Variates for Outcome Selection Bias

We now present the constructions of new control variates to improve the efficiency of ATE estimates when the data suffer from outcome selection bias. Such selection bias occurs frequently in case-control studies in epidemiology (Rothman et al., 2008; Robins, 2001; Robins et al., 2000), and in recommender systems as a problem with implicit feedback (Wang et al., 2016; Schnabel et al., 2016; Wang et al., 2020). However, current methodology for utilizing such selection-biased data for causal inference has been limited. While we focus on selection bias for the rest of the paper, we discuss applications of the control variates strategy to other data settings in Section 7.

We instantiate the main data $\mathcal{O}_1$ as observational data that suffers from selection bias on the outcome, from which the ATE is unidentifiable. Under outcome selection bias, we show that we are able to obtain consistent odds ratio estimates in various models, which can then be used to construct control variates across all datasets.

**Outcome selection bias.** Outcome selection bias is induced by preferential selection of units based on the outcome. To illuminate the nature of this bias, consider the model of Figure 1, in which $S \in \{0, 1\}$ represents the selection mechanism: $S = 1$ means presence in the sample, and $S = 0$ means absence. Recall that $X$ represents the pre-treatment covariates, $Z$ represents a binary treatment, and $Y$ represents a binary outcome.

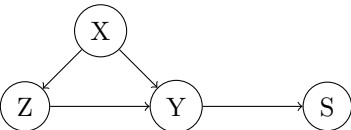

Figure 1: Causal graph for dataset $\mathcal{O}_1$ where there is a selection bias that depends on the outcome.

Under the existence of such selection bias on $Y$, the ATE is in general not identifiable, even if we assume that there are no unobserved confounding between $Z$ and $Y$. Thus, we consider the main data $\mathcal{O}_1$ to be a dataset suffering from outcome selection bias, and the validation data $\mathcal{O}_2$ to be unbiased (e.g., obtained from small-scale randomized trials).

**Variance reduction with the odds ratios.** Eq. 2 applies to any estimand $\psi$, as long as we are able to obtain *consistent* estimators of it from both datasets. Therefore, much of the difficulty in applying Eq. 2 lies in establishing a proper choice of the control variate $\psi$. We show that when $\mathcal{O}_1$ suffers from outcome selection bias, we can use the conditional odds ratios (OR) as the choice of $\psi$ since they are robust to outcome selection bias under a variety of data generating processes. Eq. 2 further shows that the strength of the variance reduction depends on the degree of correlation between $\psi$ and the ATE. The correlation between the odds ratio and the ATE has been explored empirically in the literature on case control studies and clinical trials (Ranganathan et al., 2015; Kim, 2017; Holmberg M. J., 2020). In particular, Ranganathan et al. (2015) note that for certain events, "risk approximates odds," so the ATE and odds ratio should be strongly correlated. In particular, under the varying coefficient logistic model, we derived such a relationship explicitly in Appendix C.3.

**Definition 4.1.** *The conditional odds ratio (OR) between a binary treatment $Z$ and a binary outcome $Y$ conditioned on covariates $X$ is ($x$ denotes $X = x$):*

$$OR(x) = \frac{P(Y(1) = 1|x)P(Y(0) = 0|x)}{P(Y(1) = 0|x)P(Y(0) = 1|x)}.$$

Under Assumption 2.1, we have $P(Y(1) = 1|x) = P(Y(1) = 1|Z = 1, x) = P(Y = 1|Z = 1, x)$. Therefore, we can rewrite $\mathrm{OR}(x)$ in Definition 4.1 as:

$$\mathrm{OR}(x) = \frac{P(Y = 1|Z = 1, x)P(Y = 0|Z = 0, x)}{P(Y = 0|Z = 1, x)P(Y = 1|Z = 0, x)}.$$

Proposition 4.1 further allows us to directly estimate the conditional odds ratios empirically with finite samples, even under selection bias.

**Proposition 4.1.** *(Proof in Appendix C) If the selection $S$ depends solely on $Y$ (as in Figure 1), then the conditional odds ratio is transportable and given by:*

$$OR(x) = \frac{P(Y = 1|S = 1, Z = 1, x)P(Y = 0|S = 1, Z = 0, x)}{P(Y = 0|S = 1, Z = 1, x)P(Y = 1|S = 1, Z = 0, x)}.$$

Proposition 4.1 guarantees that identifiability of the OR does not rely on any further modeling assumption (see also Didelez et al., 2010; Jiang & Ding, 2017), as long as the selection bias is on the outcome $Y$. This is a special case of (Bareinboim & Pearl, 2012), which provided other classes of causal structure models under which the identifiability of OR holds. In particular, the validation dataset and the observational dataset does not need to share exactly the same data generation process as long as the odds ratio is unchanged. Therefore, to construct control variates, it is sufficient to derive consistent estimators of the odds ratio from the datasets $\mathcal{O}_1$ and $\mathcal{O}_2$. Let $\widehat{\mathrm{OR}}_1(x)$ and $\widehat{\mathrm{OR}}_2(x)$ denote consistent estimators for $\mathrm{OR}(x)$ obtained from the datasets $\mathcal{O}_1$ and $\mathcal{O}_2$, respectively. For a set of covariate values $\{x_1, \cdots, x_k\}$, one possible control variate construction is to take $\psi = \left(\mathrm{OR}(x_1), \cdots, \mathrm{OR}(x_k)\right)^\top$. Then $\widehat{\psi}_1 = \left(\widehat{\mathrm{OR}}_1(x_1), \cdots, \widehat{\mathrm{OR}}_1(x_k)\right)^\top, \widehat{\psi}_2 = \left(\widehat{\mathrm{OR}}_2(x_1), \cdots, \widehat{\mathrm{OR}}_2(x_k)\right)^\top.$ Substituting these into Eq. 2 gives the new ATE estimator with control variates.

Although the standard odds ratios are defined for binary $Y$, the above proposition can be easily extended to categorical outcomes. In particular, we can show that $P(Y = y \mid S = 1, Z = 1, x) = P(Y = y \mid Z = 1, x)$, for $y \in \{1, \cdots K\}$ categories. We provide the proof for this in Appendix C.

When $X$ is continuous or there are too many discrete values of $X$, we can reduce the large number of conditional odds ratios $\mathrm{OR}(x)$ down to a manageable control variate by integrating $\mathrm{OR}(x)$ over a common distribution $F(x)$: let $\psi = \int \mathrm{OR}(x)F(\mathrm{d}x)$. Then $\widehat{\psi}_1 = \int \widehat{\mathrm{OR}}_1(x)F(\mathrm{d}x)$ and $\widehat{\psi}_2 = \int \widehat{\mathrm{OR}}_2(x)F(\mathrm{d}x)$.

## 4.1 Estimating the OR under selection bias

The main difficulty in applying Eq. 2 is to find consistent estimators for constructing the control variates. We demonstrate that this can be achieved by estimating the conditional odds ratios parametrically using a logistic model with varying coefficients, or non-parametrically using kernel smoothing. For the consistency analysis of estimators for ATE and the odds ratios, we assume that the data points in $\mathcal{O}_1$ and $\mathcal{O}_2$ *before* selection bias are IID samples from the same underlying population for $X, Z, Y$.

**Logistic outcome model.** One approach to estimating the odds ratio $\mathrm{OR}(x)$ uses a logistic model with varying coefficients (Cleveland, 1991) to parameterize the outcome distribution:

$$P(Y = 1|Z = z, x) = \frac{e^{\beta_0^x + \beta_1^x z}}{1 + e^{\beta_0^x + \beta_1^x z}}. \tag{4}$$

Here, $\beta_0^x, \beta_1^x$ are coefficients that depend on the covariates $x$. If $X$ is discrete and finite, then there would be a discrete and finite number of parameters $\beta_0^x, \beta_1^x$. Otherwise, $\beta_0^x$ and $\beta_1^x$ can be viewed as functions of $x$.

If the data is truly generated by the outcome model defined in Eq. 4, then Theorem 4.2 below shows that selection bias on the outcome will not change the coefficient $\beta_1^x$ across $\mathcal{O}_1$ and $\mathcal{O}_2$. Furthermore, $\beta_1^x$ is the only parameter needed to compute the conditional odds ratio $\mathrm{OR}(x)$. Thus, any consistent estimates of $\beta_1^x$ for both $\mathcal{O}_1$ and $\mathcal{O}_2$ would provide consistent estimates of the conditional odds ratio that are robust to selection bias.

**Theorem 4.2.** *(Proof in Appendix C) If the selection $S$ depends solely on $Y$ (as in Figure 1) and $P(Y = 1|Z = z, X = x)$ follows the logistic model in 4, then $P(Y = 1|Z = z, X = x, S = 1)$ also follows a logistic model, with the same coefficient $\beta_1^x$ on $Z$ as the logistic model for $P(Y = 1|Z = z, X = x)$ for each covariate value $x$. Moreover, the conditional odds ratio is $OR(x) = e^{\beta_1^x}$.*

Theorem 4.2 details the general transportability of the conditional odds ratios that is shown in Proposition 4.1 in the logistic outcome model setting. Specifically, Theorem 4.2 extends Prentice & Pyke (1979); see also Agresti (2015). By Theorem 4.2, we need only compute consistent estimators $\widehat{\beta}_{1,\mathcal{O}_1}^x$ and $\widehat{\beta}_{1,\mathcal{O}_2}^x$ of $\beta_1^x$ from $\mathcal{O}_1$ and $\mathcal{O}_2$ to produce consistent estimators of the true underlying conditional odds ratio $OR(x)$, $\widehat{OR}_1(x) = e^{\widehat{\beta}_{1,\mathcal{O}_1}^x}, \widehat{OR}_2(x) = e^{\widehat{\beta}_{1,\mathcal{O}_2}^x}$.

When $X$ is discrete, we can obtain such consistent estimators $\widehat{\beta}_{1,\mathcal{O}_1}^x$ and $\widehat{\beta}_{1,\mathcal{O}_2}^x$ by stratifying the data on $X$ and performing logistic regression within each stratum. Let $\widehat{\beta}_{1,\mathcal{O}_2}^x$ be the maximum likelihood estimator for $\beta_{1,\mathcal{O}_2}^x$ in the stratum (or subset of data) with $X = x$ from $\mathcal{O}_2$ (and $\widehat{\beta}_{1,\mathcal{O}_1}^x$ be the same for $\mathcal{O}_1$). These maximum likelihood estimators are consistent estimators for the true $\beta_1^x$.

For continuous $X$, producing a theoretically consistent estimator for $\beta_0^x, \beta_1^x$ is more challenging. One technique is to assume parametric models for the functions $\beta_0^x = f_0(x, \theta_0)$, $\beta_1^x = f_1(x, \theta_1)$. For example, if these functions are linear (i.e., $f_0(x, \theta_0) = \theta_0^\top x$, $f_1(x, \theta_1) = \theta_1^\top x$), then the problem of estimating $\beta_1^x$ reduces to maximum likelihood estimation of $\theta_1$ over a logistic model: $P(Y = 1|Z = z, x) = e^{\theta_0^\top x + \theta_1^\top xz}/(1 + e^{\theta_0^\top x + \theta_1^\top xz})$. We may also allow $f_0(x, \theta_0), f_1(x, \theta_1)$ to take more general functional forms, such as neural networks. Depending on the complexity of the functions, it becomes more challenging to guarantee asymptotic consistency theoretically and obtain a rate of convergence to the true $\beta_1^x$. However, such methods may still work well in practice. We explore their empirical performance in Section 6.

**Kernel smoothing.** When $X$ is continuous, we can also estimate the odds ratio using kernel smoothing without making any parametric assumptions on the exact outcome model or functional form of $\beta_1^x$. First, notice that $OR(x) = \frac{\mathbb{E}[YZ|x] \cdot \mathbb{E}[(1-Y)(1-Z)|x]}{\mathbb{E}[Y(1-Z)|x] \cdot \mathbb{E}[(1-Y)Z|x]}$. Further, by Proposition 4.1,

$$OR(x) = \frac{\mathbb{E}[YZ|S = 1, x] \cdot \mathbb{E}[(1-Y)(1-Z)|S = 1, x]}{\mathbb{E}[Y(1-Z)|S = 1, x] \cdot \mathbb{E}[(1-Y)Z|S = 1, x]}.$$

Therefore, estimating $OR(x)$ is equivalent to estimating $\mathbb{E}[W|x]$ and $\mathbb{E}[W|S = 1, x]$ from $\mathcal{O}_1$ and $\mathcal{O}_2$, respectively, where $W \in \{YZ, (1-Y)(1-Z), Y(1-Z), Z(1-Y)\}$. Choose a kernel function $K(\cdot)$ and the bandwidth $\lambda$. Given a dataset with $n$ data points $(X_i, Y_i, Z_i)_{i=1}^n$, for a random variable $W \in \{YZ, (1-Y)(1-Z), Y(1-Z), Z(1-Y)\}$, $\hat{\mathbb{E}}[W|x] = \sum_{i=1}^N K(\frac{x-X_i}{\lambda})W_i / \sum_{i=1}^N K(\frac{x-X_i}{\lambda})$. Therefore, the kernel estimator is:

$$\widehat{OR}(x) = \frac{\sum_{i=1}^N K(\frac{x-X_i}{\lambda})Y_i Z_i \sum_{i=1}^N K(\frac{x-X_i}{\lambda})(1-Y_i)(1-Z_i)}{\sum_{i=1}^N K(\frac{x-X_i}{\lambda})Y_i(1-Z_i)K(\frac{x-X_i}{\lambda})(1-Y_i)Z_i}.$$

This estimator is consistent under selection bias on the outcome as shown by Proposition 4.1. Unlike the parametric estimators using the MLE, we note that the asymptotic convergence of the kernel estimator depends on the bandwidth $\lambda$ and the dimensionality $d$. We provide further analysis of this convergence for the odds ratio in Appendix A.

## 5 Simulation Experiments

We first demonstrate the finite-sample performance of estimators with and without the proposed control variates in a simulation study. We simulate an observational dataset with confounding from $X$ using a logistic model adapted from Zhang (2009).

### 5.1 Data generation

We generate the dataset $\mathcal{O}_2$ by sampling $n_2$ samples from the following data-generating process. Let $X \in \mathbb{R}^2$ have two components $X_1, X_2$, which are IID Bernoulli$(p = 0.5)$. Given $X$, the treatment assignment $Z$

is distributed as $P(Z = 1|X = x) = e^{a_0 + a_1^\top x}/(1 + e^{a_0 + a_1^\top x})$. As done by Zhang (2009), the outcome $Y$ is generated from a logistic model with an interaction term between $X$ and $Z$ parameterized by $\{\beta_i\}_{i=0}^3$:

$$P(Y = 1|Z = z, x) = \frac{e^{\beta_0 + \beta_1 z + \beta_2^\top x + \beta_3^\top xz}}{1 + e^{\beta_0 + \beta_1 z + \beta_2^\top x + \beta_3^\top xz}}. \tag{5}$$

We specifically set $\beta_3 \neq 0$ so that the conditional odds ratio varies as a function of $x$. Full details with exact parameters settings are given in Appendix D.1. To generate $\mathcal{O}_1$, we first draw $(Z_i, X_i, Y_i)_{i=1}^N$ samples from the same data-generating process as $\mathcal{O}_2$, and include each sample $(Z_i, X_i, Y_i)$ in $\mathcal{O}_1$ with probabilities $P(S_i = 1|Y_i = 1) = 0.9$ and $P(S_i = 1|Y_i = 0) = 0.1$. This simulates selection bias in favor of positive outcomes as happens in case-control studies in practice.

## 5.2 Estimating the ATE and the control variate

To obtain an estimate $\widehat{\tau}_2$ of the ATE from $\mathcal{O}_2$, we use a parametric imputation estimator. Denote the coefficient and intercept resulting from logistic regression of $Y$ on $Z$ for stratum $X = x$ as $\widehat{\beta}_1^x$ and $\widehat{\beta}_0^x$. The regression imputation estimator of the ATE is given by:

$$\widehat{\tau}_2 = n_2^{-1} \sum_{i=1}^{n_2} \left\{ \frac{e^{\hat{\beta}_0^{X_i} + \hat{\beta}_1^{X_i}}}{1 + e^{\hat{\beta}_0^{X_i} + \hat{\beta}_1^{X_i}}} - \frac{e^{\hat{\beta}_0^{X_i}}}{1 + e^{\hat{\beta}_0^{X_i}}} \right\}. \tag{6}$$

This logistic regression model is well specified as it coincides with the true data-generating model.

To estimate the conditional odds ratio, we perform logistic regression of $Y$ on $Z$ on each stratum with $X = x$ to obtain estimates of $\beta_1^x$ from both $\mathcal{O}_1$ and $\mathcal{O}_2$, which produces estimates $\widehat{\mathrm{OR}}_1(x), \widehat{\mathrm{OR}}_2(x)$ as described in Section 4.1. To compute the proposed control variates estimator $\widehat{\tau}_{\mathrm{CV}}$ (Eq. 2), we ran $B = 100$ bootstrap replicates to estimate the co-variances $\Gamma$ and $V$, which we use to estimate the optimal control variate coefficient $\widehat{\Gamma}^\top \widehat{V}^{-1}$.

## 5.3 Finite-sample experiment scenarios

We consider three scenarios to analyze the finite-sample performance of the proposed estimators.

**Scenario 1:** We vary the size of the observational dataset, $n_2$, while keeping a constant ratio for the size of the observational dataset relative to the size of the selection biased dataset: $n_2/n_1 = 1/10$. This illustrates the simple asymptotic performance of the estimators as the sample sizes increase without changing the proportional sizes of the two datasets relative to each other.

**Scenario 2:** We vary the size of the observational dataset, $n_2$, while keeping the size of the selection biased dataset constant and relatively large: $n_1 = 10000$. This illustrates the scenario when a practitioner has access to a large fixed amount of case-control data with selection bias ($\mathcal{O}_1$), and must decide how much observational or experimental data to collect with identifiable ATE ($\mathcal{O}_2$).

**Scenario 3:** We vary the size of the selection bias dataset, $n_1$, while keeping the size of the observational dataset constant and relatively small: $n_2 = 1000$. This illustrates the relative utility of including more selection biased samples to estimate the control variate. While the ratio $n_2/n_1 = 1/10$ is fixed in Scenario 1, in Scenario 3 we consider the effect of varying that ratio for a fixed observational dataset size, $n_2$.

## 5.4 Results

Figure 2 compares the variance for the ATE estimator $\widehat{\tau}_2$ with the variance of the ATE estimator with control variates $\widehat{\tau}_{\mathrm{CV}}$ for the three different finite-sample scenarios varying $n_1$ and $n_2$. The variances of these estimators are measured over $B = 100$ bootstrap replicates. Throughout all three scenarios, the estimator with control variates $\widehat{\tau}_{\mathrm{CV}}$ had significantly reduced variance compared to $\widehat{\tau}_2$ alone. However, the impact of increasing $n_1$ and $n_2$ varies, with $n_2$ mattering much more for improving the variance. Results for Scenario 1 and Scenario 2 (Figure 2 *left* and *middle*) show that the variance of $\widehat{\tau}_2$ and $\widehat{\tau}_{\mathrm{CV}}$ both decrease significantly as $n_2$ increases, even if the ratio $n_2/n_1$ is not necessarily fixed. However, in Scenario 3 (Figure 2 *right*),

we observe that when there is a limited fixed amount of observational data $n_2$, increasing the amount of selection-biased data does not seem to significantly improve the variance of the estimator with control variates, $\hat{\tau}_{\text{CV}}$. We further report the bias of each estimator over the bootstrap replicates in Figure 3. In general, the bias decreases as $n_2$ increases, and is not significantly different with or without control variates.

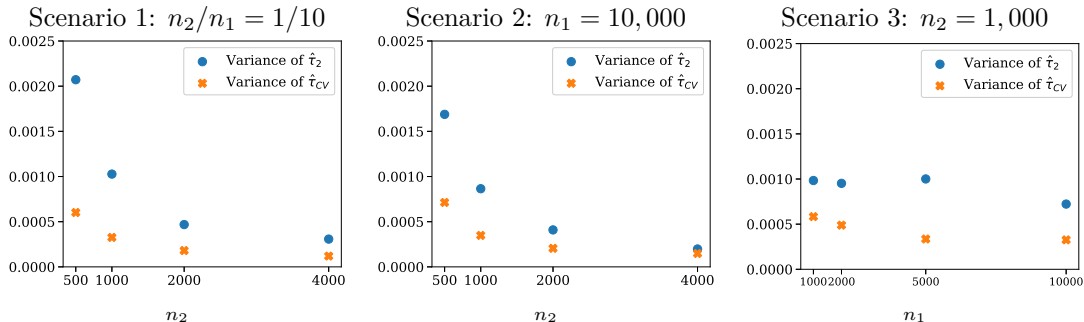

Figure 2: Comparisons of variance for $\hat{\tau}_2$ and $\hat{\tau}_{\text{CV}}$ over 100 bootstrap replicates. *Lower is better.*

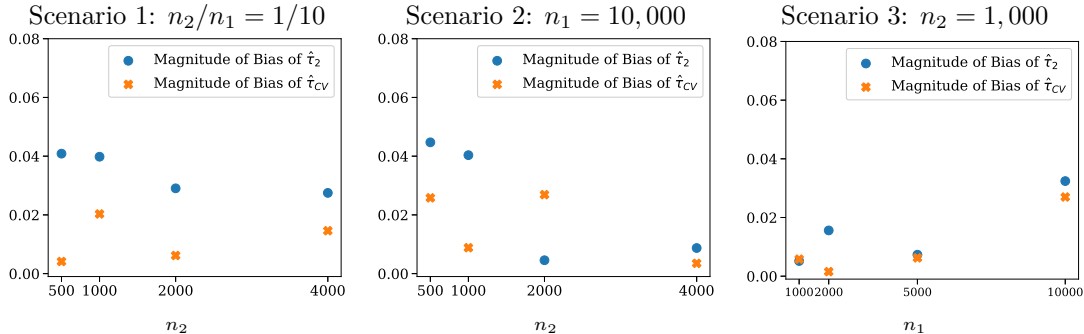

Figure 3: Comparisons of bias for $\hat{\tau}_2$ and $\hat{\tau}_{\text{CV}}$ over 100 bootstrap replicates. The magnitude of bias reported is the absolute value of the difference between the average value of the estimator over the boostrap replicates and the true ATE. *Lower is better.*

### 5.4.1 Comparisons to baselines and related methods

**Comparison to Correa et al. (2019).** While Correa et al. (2018; 2019) proved that the causal effect is not identifiable from $\mathcal{O}_1$ alone in this setting, Correa et al. (2019) showed that when given access to an unbiased data distribution, the causal effect is identifiable. They also outline an algorithm for combining a biased and unbiased dataset; however, their algorithm assumes access to full data distributions and does not provide any finite sample estimators. Since our work focuses on finite sample estimators with samples from datasets $\mathcal{O}_1$ and $\mathcal{O}_2$, it is not directly comparable to the algorithm from Correa et al. (2019), but finite sample estimators inspired by Correa et al. (2019) would be interesting to compare to our proposed finite sample estimator $\hat{\tau}_{\text{CV}}$.

The difficulty of creating finite sample estimators from Correa et al. (2019) depends on the complexity of the data. For low-dimensional discrete covariates $X$, we can directly compute finite sample estimates of the required joint probability distributions. Since this simulation experiment has low dimensional $X$, we create for comparison purposes a finite sample ATE estimator based on the methodology from Correa et al. (2019), which we denote with $\hat{\tau}_{[\text{CTB'19}]}$. The specific details of this estimator can be found in Appendix D.2.

For high dimensional and continuous covariates X, finite sample estimation methods are not as straightforward and as far as we know have not been suggested for Correa et al. (2019). Thus, we do not include this baseline for the real data experiments.

**Additional baseline estimators.** In addition to the basic regression imputation estimator $\widehat{\tau}_2$, the control variate estimator $\widehat{\tau}_{\text{CV}}$, and the estimator based on Correa et al. (2019) $\widehat{\tau}_{[\text{CTB'19}]}$, we include two additional baselines for comparison: *(i)* a regression imputation estimator using only the biased dataset $\mathcal{O}_1$ (denoted $\widehat{\tau}_1$), and *(ii)* a regression imputation estimator using a concatenation of $\mathcal{O}_1$ and $\mathcal{O}_2$ (denoted $\widehat{\tau}_{1\&2}$).

Figure 4 illustrates the full spreads of each estimator over 100 bootstrap replicates relative to the true ATE. Overall, $\widehat{\tau}_{[\text{CTB'19}]}$ performs very similarly to the proposed $\widehat{\tau}_{\text{CV}}$ in Scenarios 1 and 2. Scenario 3 shows worse performance for $\widehat{\tau}_{[\text{CTB'19}]}$ for low $n_1$, since the methodology by Correa et al. (2019) only utilizes unbiased data from the subset of variables that directly affect $S$ (which in our case is only the variable $Y$), and gets all other information from the biased data. With less biased data, $\widehat{\tau}_{\text{CV}}$ performs worse. This effect may be mitigated with more sophisticated finite sample methods, which would make for interesting future study.

The estimator $\widehat{\tau}_{1\&2}$ applies a naive combination of the datasets, and there is no theoretical guarantee that this estimator will be unbiased. In fact, results in Figure 4 show that the bias of $\widehat{\tau}_{1\&2}$ is significantly higher than that of $\widehat{\tau}_{\text{CV}}$.

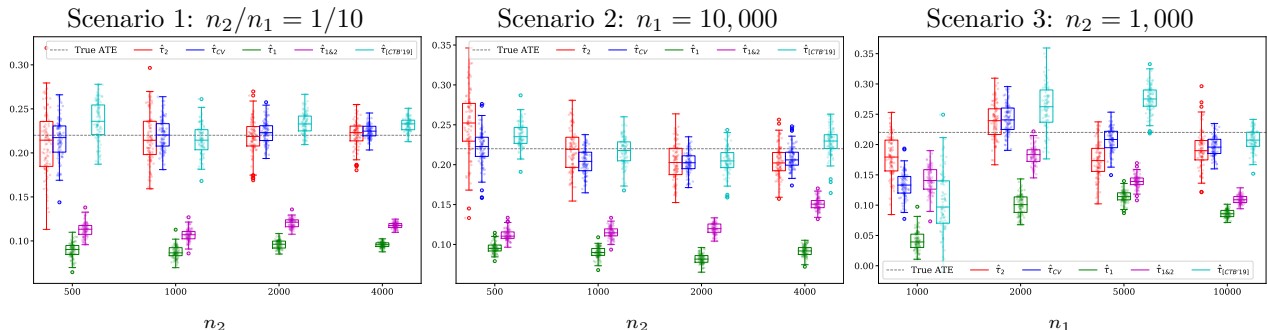

Figure 4: Comparison of ATE estimators $\widehat{\tau}_2$ and $\widehat{\tau}_{\text{CV}}$ with additional baselines $\widehat{\tau}_1$, the naive dataset combination $\widehat{\tau}_{1\&2}$, and $\widehat{\tau}_{[\text{CTB'19}]}$ (shown from left to right for each dataset size). Standard box plots over 100 bootstrap replicates show error bands given by the 25th percentile, 75th percentile, min, and max (with outliers circled). The true ATE is shown with the grey dashed line.

# 6 Real-Data Case Studies

We further evaluated the performance of the proposed control variates on two case studies with public datasets. All code will be made publicly available.

**Case study 1: flu shot encouragement with selection bias from case-control studies.** We consider a flu shot encouragement experiment dataset that has been repeatedly studied in the causal inference literature (McDonald et al., 1992; Hirano et al., 2000; Ding & Lu, 2017). This data is for 2,861 patients collected from an encouragement experiment in which participating physicians were assigned treatments $Z$ uniformly at random, where $Z = 1$ indicates that the patient's physician was sent a letter encouraging them to vaccinate their patients ($Z = 0$ otherwise). The binary outcome $Y$ is whether the patient was hospitalized for flu-related reasons the following winter. The dataset contains eight additional covariates $X$, one of which is a continuous *age* variable, and seven of which are binary indicators of prior patient medical conditions (e.g., history of heart disease). We do not consider the intermediate variable of whether the patient received a flu shot.

We set $\mathcal{O}_2$ to be this original dataset. For the second dataset $\mathcal{O}_1$, we consider a realistic scenario where an additional observational dataset exists consisting mostly of patients who have already been hospitalized for flu-related reasons. Observational studies consisting mostly of positive outcomes are common in epidemiology as case-control studies. Such a dataset may be easier to collect than the original encouragement experiment, since such data may already be available from hospitals without setting up an explicit controlled experiment. We simulate $\mathcal{O}_1$ by training a logistic model with interaction terms parameterized by Eq. 5 on the original dataset $\mathcal{O}_2$, and generating samples according the fitted distribution (details in Appendix E). We assume that the "true" ATE is given by this model.

**Case study 2: spam email detection with selection bias from implicit feedback.** For a second case study, we use a dataset constructed for the Atlantic Causal Inference Conference (ACIC) 2019 Data Challenge based on the Spambase dataset for spam email detection from UCI (Gruber et al., 2019; Dua & Graff, 2017). The dataset consists of emails with outcome of interest $Y$ being whether a user marked the email as spam or not. The treatment $Z$ is whether or not the email contains more than a given threshold of capital letters (the threshold is computed by a mean over the full dataset). There are 22 continuous covariates $X$ which are word frequencies given as percentages. The ACIC competition does not use the original data from UCI directly, but instead generates modified versions using pre-specified data generating processes with known true ATE. We generate $\mathcal{O}_2$ using ACIC's data generating process, for which we provide more details in Appendix E.

For $\mathcal{O}_1$, we generate data from the same data generating process and apply selection bias $P(S = 1|Y = 1) = 0.9$ and $P(S = 1|Y = 0) = 0.1$ to produce $n_1 = 30,000$ examples. This simulates a practical scenario where a user marking an email as spam constitutes explicit feedback, but a user *not* taking action to mark an email as spam constitutes *implicit* feedback. This implicit feedback is unreliable, since if a user does not mark an email as spam, there is no guarantee that the user actually even read the email in full. This implicit feedback problem and resulting selection bias have been repeatedly identified as a fundamental challenge in recommender systems (Wang et al., 2016; Schnabel et al., 2016; Wang et al., 2020). Disregarding the unreliable implicit feedback, the resulting dataset containing only explicit feedback is subject to selection bias on the outcome $Y = 1$ of being marked as spam, rendering the ATE non-identifiable. In our experiment, $\mathcal{O}_2$ is assumed to be a small curated dataset without the implicit feedback problem, which may be constructed by, e.g., asking users to explicitly mark emails as "not spam." We conduct two experiments to illustrate the effects of the size of this curated dataset, with $n_2 = 3,000$ and a larger $n_2 = 10,000$.

## 6.1 Estimators and implementation

For continuous $X$, we estimate the conditional odds ratio for a finite set of values $\mathcal{X}$ taken from $\mathcal{O}_2$, and set the control variate $\psi$ to be an average over all of these conditional odds ratios. Furthermore, for these datasets with continuous covariates $X$, we found that using the log conditional odds ratio was more effective as a control variate as it had lower variance for extreme values of $X$. We report results with the control variate estimand $\psi = |\mathcal{X}|^{-1} \sum_{x \in \mathcal{X}} \log \mathrm{OR}(x)$. To estimate the ATE $\widehat{\tau}_2$ and the odds ratios for the control variates, we apply three different methods:

**Logistic model with interaction:** We first apply a logistic model with an interaction term between $X$ and $Z$ (Eq. 5) to estimate both the ATE and conditional odds ratios (details in Appendix E). Since this is the same model used to generate the flu shot encouragement dataset $\mathcal{O}_1$, there is no model misspecification when using this estimator for case study 1. We set $\mathcal{X}$ to be all $X_i$ in $\mathcal{O}_2$.

**Neural network:** To allow for more flexibility, we use a neural network to estimate a logistic outcome model with varying coefficients (Eq. 4), where $\beta_0^x = f_0(x; \theta)$, $\beta_1^x = f_1(x; \theta)$ are outputs of the neural network with parameters $\theta$. The optimization objective is the logistic loss on the final outcome prediction, and we choose the neural network architecture using five-fold cross validation (more details in Appendix E). We estimate both the ATE and the odds ratio using this neural network varying coefficient model. We only report results with the neural network on the spam email dataset since the flu dataset contains a small number of mostly binary covariates, and the added flexibility from the neural network does not provide much additional benefit. We set $\mathcal{X}$ to be all $X_i$ in $\mathcal{O}_2$.

**Kernel smoothing:** As a third technique, we estimate the ATE using the logistic model in Eq. 5, but apply kernel smoothing to estimate the odds ratios for the control variate (as in Section 4.1). This non-parametric estimate sidesteps any problems of model misspecification when estimating the odds ratio. We set $\mathcal{X}$ to be a random sample of 50 values of $X_i$ from $\mathcal{O}_2$. As in the simulation study, we ran $B = 300$ bootstrap replicates to estimate the variance of the ATE estimate and the covariance between the ATE estimate and the OR control variates, which we use to compute the optimal control variate coefficient $\widehat{\Gamma}^\top \widehat{V}^{-1}$.

## 6.2 Results

Tables 1 and 2 report the variances of the ATE estimators with and without the proposed control variates, where the variance is computed over $B = 300$ bootstrap replicates. We also include the bias results in Tables 3 and 4. For both case studies, the variances of all estimators is reduced with the control variates. However, the amount that the variance is reduced varies significantly, which we discuss below.

**Comparison of estimators.** For both case studies, kernel smoothing with control variates ($\hat{\tau}_{\text{CV}}$) achieves the lowest variance among all estimators. Furthermore, both case studies exhibit significant improvement when control variates are introduced with the kernel smoothing estimator ($\hat{\tau}_{\text{CV}}$), with a $\sim 77\%$ variance decrease in case study 1 and a $\sim 21\%$ variance decrease in case study 2. Interestingly, the variance reduction of the neural network is higher than the logistic model when there are more samples $n_2$ (5.940% vs. 0.874% in Table 2), which suggests possible overfitting of the neural network control variates when $n_2$ is too small.

**Comparison of case studies and model misspecification.** Case study 2 exhibits a smaller improvement than case study 1, which we hypothesize is due to two major differences between the case studies. First, case study 1 only contains 1 continuous covariate (age) and 7 binary covariates, whereas case study 2 is significantly more complex with 22 continuous covariates. Second, the true ATE in case study 1 is given by a logistic model with interaction terms, so an estimator based on the logistic model is well specified. In contrast, the ATE estimators for case study 2 do not reflect the true underlying data generating process and are subject to model misspecification. Kernel smoothing performs the best in case study 2 since this non-parametric estimation method mitigates the issue of model misspecification. The lack of variance reduction for the parametric logistic and neural network estimators in case study 2 suggests that variance reduction from control variates may not work well for misspecified or unsuitable models in realistic applications when the true data generating process is not known.

Finally, variance reduction sometimes came at a cost of higher bias, which was more pronounced in case study 2 (Tables 3 and 4 in the Appendix). This speaks to the care that is needed in managing the bias/variance tradeoff via control variates when there is possible model misspecification.

Table 1: Variances for case study 1: flu shot encouragement data with $n_1 = 10,000$ and $n_2 = 2,861$. For case study 1, we do not include the neural network estimator since the covariates $X$ consist of only 1 continuous feature (age) and 7 other binary features, making for a relatively simple input space. Furthermore, since the "true" ATE in case study 1 is already given by a logistic model with interaction terms, there is no model misspecification that would necessitate a more flexible model like a neural network. The estimator with the lowest variance is in bold.

| Model type | Var (% diff) | Var $\hat{\tau}_2$ | Var $\hat{\tau}_{\text{CV}}$ |
|---|---|---|---|
| Logistic | 71.160% | $1.023 \times 10^{-4}$ | $2.952 \times 10^{-5}$ |
| Kernel | 77.599% | $1.072 \times 10^{-4}$ | $\mathbf{2.400 \times 10^{-5}}$ |

Table 2: Variances for case study 2: spam email detection data with $n_1 = 30,000$. We provide results for a smaller validation dataset $n_2 = 3,000$ on the left, and results for a larger validation dataset $n_2 = 10,000$ on the right. The estimator with the lowest variance is in bold.

| Model type | $n_2 = 3,000$ | | | $n_2 = 10,000$ | | |
|---|---|---|---|---|---|---|
| | Var (% diff) | Var $\hat{\tau}_2$ | Var $\hat{\tau}_{\text{CV}}$ | Var (% diff) | Var $\hat{\tau}_2$ | Var $\hat{\tau}_{\text{CV}}$ |
| Logistic | 3.522% | $4.576 \times 10^{-4}$ | $4.415 \times 10^{-4}$ | 0.874% | $1.443 \times 10^{-4}$ | $1.430 \times 10^{-4}$ |
| Kernel | 21.231% | $4.309 \times 10^{-4}$ | $\mathbf{3.394 \times 10^{-4}}$ | 23.134% | $1.351 \times 10^{-4}$ | $\mathbf{1.038 \times 10^{-4}}$ |
| Neural Net | 0.638% | $1.074 \times 10^{-3}$ | $1.067 \times 10^{-3}$ | 5.940% | $4.762 \times 10^{-4}$ | $4.479 \times 10^{-4}$ |

# 7 Further Connections

Combining multiple data sources has the potentials of mitigating bias and improving efficiency in causal inference. This work instantiates it in a setting with multiple data sources where some of them suffer from outcome selection bias, a common complication in epidemiology with limited existing methodology. Yang & Ding (2020) study the problem of combining multiple observational data sources with possible unmeasured confounding, and propose a technique for this setting which is a special case of our control variates framework. They consider a setting with two observational datasets where one large dataset contains unmeasured confounding, while another smaller dataset contains supplementary information on the confounders. They assume the two datasets have the same confounding structure (both observed and unobserved), and construct control variates which are error-prone estimators for the ATE that are transportable across the datasets. However, their approach cannot deal with outcome selection bias.

Our control variates framework is also applicable to other data combination settings. For example, combining randomized control trials with observational studies is another practically important problem that has received significant attention. One can design other control variates by finding other quantities that are transportable between the two data sources. One example is the conditional ATE given some observed covariates. This quantity is transportable when both datasets share the same causal connection between the covariates and the outcome; only the connection between the covariates and the treatment differs. When these conditional ATEs are not identified, the corresponding error-prone estimators can be used to construct the control variate. We leave the detailed study of this setting to future work.

### Acknowledgments

We wish to acknowledge support from the Vannevar Bush Faculty Fellowship program under grant number N00014-21-1-2941, the Google PhD fellowship program, the National Science Foundation Graduate Research Fellowship Program under grant number DGE 1752814, and the National Science Foundation under grant number 1945136 and NSF-CHE-2231174.

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
