# OpenReview forum: "Multi-Source Causal Inference Using Control Variates under Outcome Selection Bias"
_TMLR — Accepted by TMLR_

### Review · Reviewer_ZitR · 2022-08-12

**Summary Of Contributions:**

This paper studies the inference of unidentifiable causal effects using multiple data sources. Specifically, this paper focuses on the average treatment effect (ATE) under selection bias which is generally unidentifiable. To address the unidentifiability problem, the authors assume multiple data sources among which there is one (perhaps small) data source that allows the identification of the ATE. Then, similar to the control-variate methodology for general variance reduction in Monte Carlo simulations, the authors propose to find control variates to improve the efficiency of ATE estimates. The authors propose to use the conditional odds ratio (OR) as the control variate since it is robust to selection bias. Several approaches are proposed to accurately estimate OR for different types of variables. Finally, Experiments using synthetic and real-world (semi-synthetic) data show that the proposed approach can reduce the variance of the ATE estimates.

**Broader Impact Concerns:**

I don’t have any broader impact concerns.

**Requested Changes:**

1. Clearly state the consistency assumption. Provide additional theoretical analysis to show If assuming a consistent structural causal model is needed or not and why.

2. Provide additional theoretical analysis or literature study to show that the control-variate methodology also applies to interventional distributions (under certain explicitly stated assumptions).

3. Explain the difference between the proof of Proposition 4.1 and the theoretical results in [1].

4. Add more baselines in the experiments.


**Strengths And Weaknesses:**

Strengths:

This paper proposes a novel direction for estimating unidentifiable causal effects. It is reasonable to assume that we could obtain a small dataset where the target causal effect is identifiable, although it may still not be the case sometimes. The choice of OR as the control variate is also clever as the estimation of OR under selection bias is well-studied in the literature. The experiments demonstrate the efficacy of the proposed method.

Weaknesses:

1. A key assumption is not explicitly stated. My understanding is the authors assume that the true ATE \tau is the same in datasets O1 and O2 so that the control-variate methodology can be applied here. However, the authors did not explicitly state this assumption. I even doubt whether it is not enough to assume the same ATE and one must assume that the structural causal model (SCM) for generating O1 and O2 is the same. Additional theoretical studies are needed to show that the proposed method is theoretically sound.

2. A related problem is that the control-variate methodology is designed for observational distributions. Does it also apply to interventional distributions like the ATE? Additional theoretical studies or literature studies are also needed.

3. Does Proposition 4.1 follow the theoretical results in [1]? If yes, please cite [1] to give them credits. If not, please explain the difference between the proof in this paper and the theoretical results in [1].

4. Bareinboim has established theoretically sound conditions and adjustment formulas for selection bias in a single data source [2,3]. It is worthy to compare the proposed method with these baselines.

[1] Bareinboim, Elias, and Judea Pearl. "Controlling selection bias in causal inference." Artificial Intelligence and Statistics. PMLR, 2012.

[2] Correa, Juan, Jin Tian, and Elias Bareinboim. "Generalized adjustment under confounding and selection biases." Proceedings of the AAAI Conference on Artificial Intelligence. Vol. 32. No. 1. 2018.

[3] Correa, Juan D., Jin Tian, and Elias Bareinboim. "Identification of causal effects in the presence of selection bias." Proceedings of the AAAI Conference on Artificial Intelligence. Vol. 33. No. 01. 2019.

---

### Review · Reviewer_j75Z · 2022-08-28

**Summary Of Contributions:**

The paper studies the problem of estimating the average treatment effect from multiple data sources. The authors focus on a setting where one dataset suffers from selection bias, and the other is of small size. They propose an algorithm that constructs control variates from the conditional odds ratio estimates from the two datasets, and uses the control variates to improve the efficiency of the original estimator of the average treatment effect. The authors further demonstrate the performance of the proposed method through synthetic data and two real-data case studies.

**Requested Changes:**

Critical to securing my recommendation for acceptance:
1. The assumptions for Theorem A.1 are not stated very clearly. For example, I believe there should be some smoothness assumptions since Taylor expansion is used in the proof. Does the order of smoothness have an impact on the asymptotic behavior of  the estimator?
2. Some typos:
(1) Page 9, fourth line of Case study 2: outcome of interest Y being whether or a user.
(2) Equations are sometimes referred to as “Eq. equation 2” but sometimes as “equation 3”. There is also “Definition equation 4.1”. It will be great if things can be made consistent.

Would strengthen the work in my view:
1. The two real-data case studies are still semi-synthetic. Ideally, I would love to see two real datasets: one subject to selection bias (O1) and the other not (O_2’). If O_2’ is large enough, we can pretend the estimator from O_2’ is the true ATE. Then we can subsample O_2’ and get a smaller O_2. I’m very interested to see then what happens if we apply the proposed method to O_1 and O_2.
2. I would also like to see more discussion on what happens if the two datasets do not have the same “data generating process”. The data generating process can be very similar but not exactly the same. How will the proposed method perform on simulations in this case? Heuristically, are we going to see higher bias with lower variance? If so, maybe the method can still be helpful in practice even when the two datasets have different “data generating processes”. In practice, is there a way to decide whether we should use the control variates or not? Thanks!


**Strengths And Weaknesses:**

Strengths
1. The paper is well-written. The proposed method is explained in a very nice way.
2. The paper addresses an interesting and important question: leveraging datasets that may suffer from selection bias to improve the efficiency of the original causal estimator.
3. The proposed method is very elegant and simple to implement.
4. The proposed method appears to have good performance both theoretically and empirically.

Weaknesses
1. The two real-data case studies are still semi-synthetic. (More details in the next section).
2. I would like to see more discussions on what happens if the two datasets do not have the same “data generating process”. (More details in the next section).

---

### Review · Reviewer_qNNR · 2022-09-02

**Summary Of Contributions:**

This paper proposes to use multiple datasets to identify ATE for a certain dataset under outcome selection bias. Authors also apply the control variate method to reduce variance of the ATE estimation. Authors show that a special control convariate for binary treatment and control -- odds ratio (OR) can be recovered from outcome selection bias. In addition, they show that when OR satisfies certain parameteric assumptions, it can be recovered.


**Requested Changes:**

1. The choice of the control variate $\phi$ to be odds ratio (OR) limits the outcome to be binary. But the control variate based estimator for ATE is applicable to continuous outcomes. I wonder if the authors can adopt another control variate which can work for continuous outcome.
2. Theorem 4.2 only works for the generalized linear model eq.(3), which is quite limited. I wonder if a more general Theorem can be derived to show OR or even more general control variate that can be transferred from the subpopulation under selection bias to the whole population.
3. It is not clear what is the condition that the multiple datasets have to satisfy to identify the ATE in a certain dataset under outcome selection bias. It seems the authors only consider two datasets, one under outcome selection bias, the other one is from randomized controlled trial, which makes the claim in abstract and introduction mismatch with what has been really done.


**Strengths And Weaknesses:**

Strength
1. This work aims to solve an extremely challenging problem, i.e., recovering ATE from outcome selection bias, from a new perspective.
2. Experiments are done on both synthetic and real datasets. Results show the effectiveness of control covariate (especially the kernel smoothing estimator).

Weakness
1. The contribution of this work is quite limited and can be much less limited if the authors replace certain design choices like: using OR as the only control covariate -- leading to the limitation that only binary outcome can be considered. Theorems only work for a certain choice of control covariate and a specific parameterization of the control covariate.
2. With the proposed method, I cannot clearly see how the proposed method finds and uses "features that are transportable across both datasets" claimed in the introduction/abstract. Instead, the transportability only comes from the OR function which can be unbiasedly estimated from the data under selection bias.

---

> ### Author Response · Authors · 2022-09-15
> **Author response to reviewer qNNR**
>
> Dear reviewer,
>
> Thank you for your thoughtful review! We hope our response answers your questions. Please note that we have also updated the PDF to reflect the revisions.
>
> ## Limitation with binary outcomes
>
> You are correct that the OR control variate is typically defined for binary outcomes. However, the main conclusion about the transportability of such control variates can indeed be extended to categorical / ordinal outcomes. Specifically, we can show that $P(Y=k \mid Z, X) = P(Y=k \mid Z, X, S=1)$, for $Y \in \{1, 2, \dots K\}$ categorical outcomes, which is the key step in proving Proposition 4.1. For continuous outcomes, one may also consider categorizing different ranges of the $Y$ values into different buckets, such that the problem reduces categorical outcomes. For example, if $Y$ is the recovery time for a certain disease, the buckets can represent ranges for short / mid / long recovery times.
>
> Prior works have also proposed generalizations of odds ratios to ordinal outcomes (Generalized Odds Ratios for Ordinal Data, Alan Agresti, Biometrica), which can be used as the control variates in that case. **We have added additional comments and references on this extension in the revision, and extended the proof for Proposition 4.1 to categorial $Y$.**
>
> ## Theorem 4.2 only works for the generalized linear model eq.(3); can a more general theorem be derived to show the transportability of OR?
>
> You are correct that Theorem 4.2 detailed the specific transportability of the OR coefficients for the generalized linear model. However, the conditional ORs are consistent across the validation dataset and the selection-biased dataset is a general conclusion, and we do have such a general statement. As shown in Proposition 4.1: $P(Y=y \mid X, Z) = P(Y=y \mid X, Z, S=1)$. Theorem 4.2 instantiated the estimation details when we posit a generalized linear model for the data generation mechanism. **We have added additional comments in the paper to clarify this point.**
>
> ## Details on the application to multiple datasets
>
> Thanks for the helpful suggestion. The proposed control variates strategy is not limited to two datasets, and can be easily extended to multiple datasets. In short, note that the control variates (see Eq(1), $\phi$) are expressed as a vector. When there is a single validation dataset and $m$ observational datasets, we can construct control variates between the validation dataset and each of the observational datasets, that is: $\hat \phi_{obs, i} - \hat \phi_{val}$ for $i = 1, \cdots, m$ observational datasets. Then, we can stack these together as a new vector of control variates.
>
> Moreover, for multiple observational data sources with different biases, different control variates can be used with different observational data sources. That is, they do not have to share the same estimand $\phi$, as long as each estimand is transportable between the validation dataset and the corresponding observational dataset. For example, [Combining Multiple Observational Data Sources to Estimate Causal Effects, Yang and Ding’20] proposed other choices of the control variates that work for other biases but not outcome selection bias.
>
> **We have added additional comments and detailed illustrations at the end of Section 3 main text for the extension to multiple datasets.**

---

### Author Response · Authors · 2022-09-15
**Message to everyone**

Dear reviewers and editor,

Thank you for reviewing this work. We present a summary of all the major changes in the revision. We have uploaded the revised paper in a PDF. For ease of reviewing, we have also attached a colored “diff” version showing the differences between the previous version and the revised version at the end of the same file.

Below we list the changes that were made in this revision.

# Changes since last submission

**As suggested by reviewer qNNR:**
- We have added additional comments and references on the extension of the odds ratio control variate beyond binary outcomes in the revision (after Proposition 4.1 and in the proof of it). We extended the proof for Proposition 4.1 to categorial $Y$.
- We have added additional comments surrounding Theorem 4.2 to clarify that it is a general conclusion that the conditional ORs are consistent across the validation dataset and the selection-biased dataset. Theorem 4.2 just details specific coefficients for the special case of a generalized linear model.
- We have added additional comments and detailed illustrations at the end of Section 3 main text for the extension to multiple datasets.

**As suggested by reviewer j75Z:**
- We have added more detailed assumptions in Appendix A for Theorem A.1.
- We have added a note to the revised version in Section 3 on the possible different data generation processes for $\mathcal{O}_1$ and $\mathcal{O}_2$

**As suggested by reviewer ZitR:**
- We have added a note in Section 3 on assumptions for the control-variate methodology.
- We have added the reference to [1] and additional clarification to the results in [1] immediately following Proposition 4.1 in the paper.
- We created a finite sample estimator based on [3] for comparison purposes, which we detail in Appendix D.2.
- We have added a comparison to [3] in Section 5.4.1 and Figure 4.
- We have also added references to [2,3] to the introduction and related work sections.

---

### Decision · Action_Editors · 2022-10-04

**Recommendation:** Accept as is

**Comment:**

There was some request by the reviewers to clarify the role of some assumptions, and potential limitations of the proposed methodology, that were addressed by the authors. This was nicely summarized by the authors in their "changes since last submission", the reviewers were satisfied with the revised version of the paper.

**Audience:**

The results in this paper are of interest for any statistician / data scientist interested in causal inference.

**Claims And Evidence:**

Non-identifiability is a chore problem of causal inference. Due to selection bias, the parameters or functionals of interest are often non-identifiable. The authors propose to take advantage of multiple sources to estimate the average treatment effect (ATE). Indeed, the ATE can be indentifiable in some datasets, and non-identifiable in others. However, the latter can be to construct control variates. These control variates can be used to reduce the variance of the estimator of the ATE in the datasets where it is actually identifiable. This claim is supported by exhaustive experiments, with both synthetic and real data.

All the reviewers agree that this is an original point of view on an important problem, and I agree with them. They also mention that the experimental results provide good support for the methodology proposed in the paper.

---

> ### Author Response · Authors · 2022-10-20
> **Camera-ready version uploaded**
>
> We thank all the reviewers and action editors for the constructive suggestions which improved the paper. We have uploaded the camera-ready version as suggested.
>
> Thank you!